# A System-on-a-Chip Implementation of a Post-Quantum Cryptography Scheme for Smart Meter Data Communications

**DOI:** 10.3390/s22197214

**Published:** 2022-09-23

**Authors:** Vinícius Lagrota Rodrigues da Costa, Julio López, Moisés Vidal Ribeiro

**Affiliations:** 1Engineering Department, Federal University of Juiz de Fora (UFJF), Juiz de Fora 36036-900, MG, Brazil; 2Institute of Computing, State University of Campinas (UNICAMP), Campinas 13083-852, SP, Brazil

**Keywords:** smart grid, smart meter, post-quantum cryptography, system-on-a-chip

## Abstract

The security of Smart Meter (SM) systems will be a challenge in the era of quantum computing because a quantum computer might exploit characteristics of well-established cryptographic schemes to reach a successful security breach. From a practical perspective, this paper focuses on the feasibility of implementing a quantum-secure lattice-based key encapsulation mechanism in a SM, hardware-constrained equipment. In this regard, the post-quantum cryptography (PQC) scheme, FrodoKEM, an alternate candidate for the National Institute for Standards and Technology (NIST) post-quantum standardization process, is implemented using a System-on-a-Chip (SoC) device in which the Field Programmable Gate Array (FPGA) component is exploited to accelerate the most time-consuming routines in this scheme. Experimental results show that the execution time to run the FrodoKEM scheme in an SoC device reduces to one-third of that obtained by the benchmark implementation (i.e., the software implementation). Also, the attained execution time and hardware resource usage of this SoC-based implementation of the FrodoKEM scheme show that lattice-based cryptography may fit into SM equipment.

## 1. Introduction

The necessity of supplying energy to an increasing and already large number of loads and dealing with their dynamics, and including renewable energy sources in the energy matrix has motivated worldwide academic and industry research efforts to put forward Smart Grid (SG) [1]. The development of SG technologies and solutions have become a priority in the electricity sector because SG is a powerful concept to ensure energy reliability and efficiency, reduce technical and non-technical losses, enable the integration of different types of renewable energy sources [2], fulfill the current and future needs and demands of all stakeholders and consumers.

In this sense, the use of the Internet of Things (IoT) in electric power systems constitutes a remarkable advance to quickly spread SG solutions worldwide and, consequently, introduces the so-called digital transformation in electric power systems [3]. Also, the upgrade of existing technologies in electric power systems offers a new perspective for accelerating the implementation of SG concepts [4,5]. In this scenario, Smart Meters (SMs) have emerged as enabling technologies for SGs because they can monitor and control the bidirectional electricity consumption and generate consumer or prosumer data. These functionalities are essential to the stakeholders in the electricity sector because it is beyond the traditional unidirectional power billing functionality [6]. It is recognized that SM data is of great value for allowing effective and dynamic energy planning to enhance electric power systems’ stability, efficiency, predictability, and reliability. Also, SM equipment allows utilities, consumers, and prosumers to access their data in near real-time.

In this scenario, security and privacy awareness have emerged to deal with security threats against the SM systems because unauthorized access to SM data can severely impact the operation of electric power systems and the life of consumers. For example, [7] showed that an attacker could exploit a flaw to change electric power system data without being noticed. This is because key generation and distribution in a cryptographic scheme may result in a security breach when SM data is transmitted through data networks [8,9]. To deal with this scenario, research efforts have been focused on privacy of information, security issues and countermeasures, see [10,11,12,13,14] and reference therein. In this sense, the research effort to secure public-key encryption schemes against classical computing attacks is worth mentioning. In particular, for practical public-key cryptosystems based on the hardness of factoring integers or computing discrete logarithms (e.g., Rivest-Shamir-Adleman (RSA) (RSA-based cryptography is much more complex due to its modular exponentiation) and Elliptic Curve Cryptography (ECC)) [15], which are used to ensure the security of SM data traveling through data networks. However, such cryptographic schemes will be easily broken when a sufficiently powerful and stable quantum computer runs Shor’s algorithm, which is a polynomial time quantum algorithm for integer factorization and discrete logarithms [16].

The design of post-quantum cryptography (PQC) schemes for SM systems is a timely issue because SM data comprise sensitive information. Furthermore, the feasibility of implementing PQC schemes in SM equipment is also important because the large-scale implementation of SM systems imposes a constraint on the cost of these devices. In this sense, the research community has been working towards implementations of PQC schemes. For instance, ref. [17] proposed a software implementation of CRYSTALS-Kyber, FrodoKEM, and NewHope using Graphics Processing Unit (GPU) seeking higher performance. On the other hand, ref. [18] presented a pure hardware implementation of the CRYSTALS-Kyber scheme using Field Programmable Gate Array (FPGA) focusing on high performance. Note that these aforementioned solutions are note suitable for SM systems because they are based on high-end platforms. It can also be found in the literature solutions using System-on-a-Chip (SoC) devices. For instance, ref. [19] discussed a RISC-V co-processor for lattice-based cryptography using a hardware to accelerate the Number Theoretic Transform (NTT) transform and hash generation using an SoC device, and [20] presented an Instruction Set Architecture (ISA) for lattice-based cryptography also based on SoC devices. Note that these SoC-based solutions, focused on a general purpose architecture for lattice-base cryptography, lacks optimizations for a specific scheme, which is of utmost importance when the final target is an SM equipment (i.e., hardware-constrained equipment).

In this regard, the following issues must be addressed to consider the effective and optimized implementation of PQC schemes for smart metering applications:To identify the most time-consuming routines, which are good candidates to be implemented in hardware, leading to better performance.To certify the PQC scheme can be implemented in hardware-constrained equipment, keeping in mind that a PQC scheme uses long keys and long ciphertext.If the implementation is viable, it must also be executed in a period that does not impact the performance of data exchanges between nodes in data networks.

In this regard, this paper focuses on the feasibility of using a PQC scheme as a Key Encapsulation Mechanism (KEM) for ensuring the security of SM data traveling through data networks (we do not aim to propose a new authentication scheme for SM equipment. When the KEM is completed, each part (SM and Meter Data Management System (MDMS)) will share a symmetric key, enabling any kind of secure communication using a symmetric cryptography [21], such as Advanced Encryption Standard (AES) using the Galois/Counter Mode, for instance). In this context, the implementation of the FrodoKEM scheme in an SoC device is detailed because this scheme demands the most extensive use of hardware resources [22] in comparison to other candidates for the National Institute for Standards and Technology (NIST) post-quantum standardization process [23]. Consequently, the successful implementation of the FrodoKEM scheme in an SM equipment, which is supposed to be hardware-constrained, constitutes a baseline for fostering the adoption of PQC in SM systems. Note that [24] presented the initial results of this investigation. The main contributions of this work are as follows:A description of a proposed hardware/software implementation of the FrodoKEM scheme for an SM equipment, which is hardware constrained, relying on an SoC device. In this sense, hardware accelerations for matrix-by-matrix multiplications and Secure Hash Algorithm and Keccak 128 (SHAKE128) hash function are presented.A performance comparison between the proposed hardware/software implementation and the software implementation (i.e., benchmark implementation) in terms of processing and execution times. Also, an analysis of the impact of the communication burden between the Advanced RISC Machine (ARM) and the FPGA when the SoC device is used to hardware accelerate matrix-by-matrix multiplications and the SHAKE128 hash function.An evaluation of the suitability of PQC schemes for hardware-constrained equipment, such as SM, which relies on SoC devices.

Based on the use of the Xilinx Zynq-7000 SoC device [25], the attained results show that the execution time of the proposed hardware/software implementation of the FrodoKEM scheme is around 1/3 compared to the benchmark implementation while being fully compliant with it, which is the official specification of the FrodoKEM scheme. In other words, a choice in favor of a PQC scheme based on the standard lattice (e.g., FrodoKEM) can efficiently run in hardware resource-constrained equipment. The attained results show that the proposed hardware/software implementation is suitable for SM equipment and, consequently, can remarkably benefit the security of SM data that are traveling through data networks. Last, we can extend this analysis for a dedicated cryptographic module presented in the MDMS server. Usually, this module is used to physically separate the cryptography part (especially private and shared keys) from the management part, adding another security layer.

The rest of this paper is organized as follows: Section 2 formulates the investigated problem; Section 3 pays attention to the background of the FrodoKEM scheme; Section 4 details the benchmark and proposed hardware/software implementations of the FrodoKEM scheme; Section 5 analyzes the hardware resource usage, execution time, and processing time of benchmark and proposed hardware/software implementations; finally, Section 6 outlines concluding remarks.

### Notation

For the sake of simplicity, we adopted the same notation used in the submitted specification of FrodoKEM [26]. Uppercase and lowercase bold letters are used for matrices and vectors, respectively. The set of all integers is indicated by Z and $Z_{q}=Z/qZ$ as quotient ring of integers modulo *q*. The inner product of two *n*-dimensional vectors **a**, **b** is represented by $\langle a,b\rangle =\sum_{i=0}^{n-1}a_{i}b_{i}$. Finally, the concatenation of two vectors is denoted by the || symbol.

## 2. Problem Formulation

Security and privacy are among the main challenges faced by SM systems as sensitive data is constantly produced and exchanged between consumers, prosumers, and electric utilities [27]. Energy consumption and other data collected from SMs are sensitive because they hide private information of consumers or prosumers from which their lifestyles and economic status can be leaked. Consequently, the concern about security breaches in SM systems and data has attracted more attention from the electricity sector. The design of privacy-preserving solutions that enable SM equipment to perform billing, operations, and value-added services is of utmost importance because SM equipment is one of the leading enablers of SG. Besides, SMs data is exchanged over Internet Protocol (IP)-based and dedicated networks to the MDMS and will be over non-dedicated ones (i.e., Internet) soon and, consequently, the risk of a security breach will be significant if an eavesdropper uses a quantum computer.

The security of SM systems involves several aspects related to consumers, prosumers, and data networks [28]. It is well-established that data networks must present protection against threats to system-level security (e.g., credential compromise, denial of service attacks), threats to services (e.g., SMs cloning, location migration), and privacy threats (e.g., interception/eavesdroppers, misuse of private data). In this context, cryptography has emerged as one of the most applied methods to protect SM data against eavesdroppers [29,30,31,32,33] when they are transmitted through data networks.

The vast majority of cryptographic schemes rely on the hardness of integer factoring assumption or the discrete logarithm problem due to its simplicity and lightweight implementation. Data networks relying on these schemes are secure nowadays because a classical computer can easily perform them; however, it is challenging for classical computers to undo them, preventing unauthorized disclosure and data breaches. This scenario will change significantly with the advent of powerful quantum computers, which running the Shor’s factoring algorithm can easily solve the problems mentioned above. Consequently, an eavesdropper, who uses a quantum computer, will be able to access sensitive data transmitted through data networks serving SM systems and bring severe consequences for stakeholders in the electricity sector. Moreover, what is more, alarming is that not only will SMs data communication be compromised in the future, but also an attacker who nowadays stores smart meter data might be able to decipher it using a quantum computer in the future. That said, recent advances in quantum computing have sparked interest in the research and development of PQC schemes, which are thought to be secure against quantum and classical computer attacks [34]. Therefore, preparing for this scenario is paramount to providing long-term security of SM data.

Figure 1 illustrates the scenario we are interested in addressing in this paper. The focus is on the security of SM data, transmitted between SM equipment and MDMS, against eavesdroppers that are capable of performing sniffer attacks. SM equipment, which is equipped with the lowest processing power in an SG, may constitute a security flaw if it is not capable of embedding a PQC scheme. Note that a PQC scheme is more complex than the current ones, such as RSA and ECC, and requires more processing power, which might be unfeasible for hardware-constrained devices such as SMs. One approach to overcome this hardware constraint is to use an SoC device, enabling a hardware/software implementation. This kind of implementation allows a remarkable acceleration in the most time-consuming and complex operations, which are hardware-implemented, while the less-complex and time-expensive tasks are software-implemented.

Based on this discussion, the following research questions arise: *Is it possible to implement a PQC scheme in a hardware-constrained equipment? Can a hardware/software implementation bring significant advantages over a software implementation? If yes, how much faster will it be? Is this implementation feasible in terms of hardware resource usage?* The following sections present answers to these research questions.

## 3. Background of the FrodoKEM Scheme

Recently, research about lattice-based cryptography has grown significantly, achieving important advances. For instance, one of the most relevant contributions was made by Lyubashevsky [35,36], who proposed a new class of lattices called ideal lattices. A lattice is considered ideal when it corresponds to an ideal in a particular algebraic structure, such as polynomial rings. This new class is the basis of the Ring-Learning With Errors (R-LWE) problem in contrast to the standard lattices, which are the basis of the Learning With Errors (LWE) problem. LWE is a mature and well-studied cryptographic primitive that relies on the hardness of the worst case of Shortest Vector Problem (SVP) in a standard lattice. On the other hand, R-LWE has an additional algebraic structure and relies on the worst case of an ideal lattice. Due to such additional algebraic, ideal lattices are more efficient than standard lattices because they need a small memory and perform better. Furthermore, while standard lattices are based mainly on matrix-vector (or matrix-matrix) multiplications, ideal lattices are based on polynomial multiplication, which considerably reduces the complexity and increases efficiency [37]. Despite this, it is hard to say a quantum attack will explore the weakness of this added algebraic structure in ideal lattices, consequently cracking the system in the future. On the other hand, standard lattices do not suffer from this potential vulnerability and can be considered a more conservative choice. Based on this analysis, the German Federal Office for Information Security (BSI) recommends the FrodoKEM scheme to protect confidential information on a long-term basis [38].

Moreover, lattice-based cryptosystems are more attractive because they are based on the worst-case hardness of lattice problems. If one succeeds in breaking the cryptography system by a slight chance, one can also solve any instance of a particular lattice problem. It brings a strong notion of security because the average-case instance is at least as hard as the worst-case instance of a related lattice problem [37].

The NIST initiated a process to solicit, evaluate, and standardize one or more quantum-secure public-key cryptography schemes [39]. More than 40% of the candidate schemes submitted to the NIST PQC standardization process were based on lattice-based cryptography, initially proposed by Ajtai [40]. Simplicity and parallelizable operations (e.g., addition, multiplication, and modular reduction) heavily influenced such submissions.

The FrodoCCS key exchange scheme [41] was designed by exchanging a little efficiency for high-security trust in the post-quantum era. Its simplicity is confirmed by applying only basic operations, such as addition and multiplication. Furthermore, its parameter adjustments are more flexible and easier to scale than ideal lattice-based schemes, such as the NewHope [42]. The latter has more restrictions as it uses the NTT algorithm for polynomial multiplication. Consequently, FrodoCCS can achieve different security levels with linear resource expenditure.

Based on FrodoCCS, the FrodoKEM scheme [26], which is a KEM, was submitted to the NIST post-quantum standardization process. It was selected for the third round of the NIST competition as one of eight alternate candidates. A benchmark implementation and a vectorized implementation for high-end Intel CPUs were posted along with its submission. There have been a few research studies on the feasibility of Frodo variants on embedded devices, such as ARMs and FPGAs [43]. However, none of these studies used an analysis based on an ARM processor and an FPGA device together. In this paper, we fill this gap by evaluating standard lattice-based cryptography and its feasibility for constraint embedded devices, with an eye on the SMs data communications. In this sense, we analyzed the benchmark implementation submitted and identified the most time-consuming operations. Subsequently, these routines are hardware accelerated using an FPGA owning a direct communication path with an ARM, which is responsible for performing the remaining tasks.

Before proceeding to the implementation section, a theoretical background review of the LWE problem, the basis of FrodoKEM, and how it applies to the FrodoKEM scheme is outlined.

### 3.1. Learning with Errors

The security of the proposed FrodoKEM relies on the hardness of the LWE problem. According to Regev [44], the LWE problem asks to recover a secret $s\in Z_{q}^{n}$ given a sequence of “approximate” random linear equations on *s*. The formal definition is as follows. Fix a size parameter n>1, a modulus q⩾2, and an error probability distribution χ on $Z_{q}$. Now, let $A_{s,\chi }$ on $Z_{q}^{n}\times Z_{q}$ be the probability distribution obtained by choosing a vector $a\in Z_{q}^{n}$ uniformly at random, choosing $\epsilon \in Z_{q}$ according to χ, and outputting (a,〈a,s〉+ϵ), where additions are performed in $Z_{q}$. Finally, it can be said that an algorithm solves LWE with modulus *q* and error distribution χ if, for any $s\in Z_{q}^{n}$, given an arbitrary number of independent samples from $A_{s,\chi }$ it outputs s (with high probability).

Therefore, the LWE problem is nothing more than a noisy system with linear equations. In general, this problem is not trivial to solve. No quantum algorithms are currently known to solve the LWE problem in polynomial time [44]. Consequently, schemes based on LWE are considered quantum-secure.

### 3.2. The Frodo Key Encapsulation Mechanism Scheme

The FrodoKEM scheme can be basically divided into three algorithms: key pair generation, encapsulation, and decapsulation [26] as described in Algorithms 1–3, respectively. A few subroutines are called by these algorithms, see [26] for more details. Briefly, the **Gen**(.) function receives as input a seed and outputs a matrix $A\in Z_{q}^{n\times n}$ which was generated using a hash function. Similarly, the **SampleMatrix**(.) function outputs a matrix sampled from the χ error probability distribution. The **Pack**(.) function transforms the received matrix into a bit string, while **Unpack**(.) function does the opposite. Finally, the **Encode**(.) function encodes bit strings as mod-*q* integer matrices. On the other hand, the **Decode**(.) function does the inverse operation. Finally, all bit string lengths (${\mathrm{len}}_{{\mathrm{seed}}_{A}}$, ${\mathrm{len}}_{{\mathrm{seed}}_{\mathrm{SE}}}$, len-, ${\mathrm{len}}_{s}$, ${\mathrm{len}}_{k}$, ${\mathrm{len}}_{\mathrm{pkh}}$, ${\mathrm{len}}_{\mathrm{ss}}$, ${\mathrm{len}}_{z}$, ${\mathrm{len}}_{\chi }$) are previously known constants; *D* is the exponent which defines the scheme modulus $q=2^{D}$; *n*, $\bar{n}$ and $\bar{m}$ are integer matrix dimensions with n≡0(mod8); and $T_{\chi }$ is the distribution table for sampling.   
**Algorithm 1:** Key pair generation**Input:** None**Output:**Public key: $pk\in {\left\{0,1\right\}}^{{\mathrm{len}}_{{\mathrm{seed}}_{A}}+D\cdot n\bar{n}},$Secret key: $sk^{\prime }\in {\left\{0,1\right\}}^{{\mathrm{len}}_{s}+{\mathrm{len}}_{{\mathrm{seed}}_{A}}+D\cdot n\bar{n}}\times Z_{q}^{n\times \bar{n}}\times {\left\{0,1\right\}}^{{\mathrm{len}}_{\mathrm{pkh}}}$**Procedure:**Choose uniformly random seeds: $s||{\mathrm{seed}}_{\mathrm{SE}}\left|\right|z\leftarrow_{\$}U\left({\left\{0,1\right\}}^{{\mathrm{len}}_{s}+{\mathrm{len}}_{{\mathrm{seed}}_{\mathrm{SE}}}+{\mathrm{len}}_{z}}\right)$Generate a pseudo-random seed: ${\mathrm{seed}}_{A}\leftarrow \mathrm{SHAKE}\left(z,{\mathrm{len}}_{{\mathrm{seed}}_{A}}\right)$Generate Matrix $A\in Z_{q}^{n\times n}$ via $A\leftarrow \mathrm{Gen}({\mathrm{seed}}_{A})$Generate pseudo-random bit string:$(r^{\left(0\right)},r^{\left(1\right)},\ldots ,r^{\left(2n\bar{n}-1\right)})\leftarrow$$\mathrm{SHAKE}(0x5F||{\mathrm{seed}}_{\mathrm{SE}},2n\bar{n}\cdot {\mathrm{len}}_{\chi })$Sample error matrix $S^{T}\leftarrow \mathrm{SampleMatrix}\left(\left(r^{\left(0\right)},r^{\left(1\right)},\ldots ,r^{\left(n\bar{n}-1\right)}\right),\bar{n},n,T_{\chi }\right)$Sample error matrix $E\leftarrow \mathrm{SampleMatrix}(\left(r^{\left(n\bar{n}\right)},r^{\left(n\bar{n}+1\right)},\ldots ,r^{\left(2n\bar{n}-1\right)}\right),n,\bar{n},T_{\chi })$Compute B←AS+ECompute b←Pack(B)Compute $\mathrm{phk}\leftarrow \mathrm{SHAKE}({\mathrm{seed}}_{A}||b,{\mathrm{len}}_{\mathrm{pkh}})$**Return:**Public key $pk\leftarrow {\mathrm{seed}}_{A}\left|\right|b$Secret key $sk^{\prime }\leftarrow \left(s|\right|{\mathrm{seed}}_{A}||b,S^{T},\mathrm{pkh})$

The main part of the key pair generation (Algorithm 1) is the calculation of the LWE sample operation B←AS+E. The matrix A is generated by a pseudo-random seed while ${\mathrm{seed}}_{A}$ is created from a uniformly random seed hashed by a function. The FrodoKEM scheme uses two hash functions: a hash based on the AES cipher and the SHAKE128 hash algorithm. The matrices E and S are sampled according to the distribution χ. Later, matrix B is packed into bit string b, and the bit string ${\mathrm{seed}}_{A}$ and b are hashed to get a hash value phk. Finally, the public key pk is composed of ${\mathrm{seed}}_{A}$ and b, while the secret key $sk^{\prime }$ is composed of s (previously uniformly random generated), ${\mathrm{seed}}_{A}$, b, $S^{T}$, and pkh.   
**Algorithm 2:** Encapsulation**Input:**Public key: $pk={\mathrm{seed}}_{A}\left|\right|b\in {\left\{0,1\right\}}^{{\mathrm{len}}_{{\mathrm{seed}}_{A}}+D\cdot n\bar{n}}$**Output:**Ciphertext $c_{1}\left|\right|c_{2}\in {\left\{0,1\right\}}^{(\bar{m}n+\bar{m}\bar{n})D}$Shared secret $\mathrm{ss}\in {\left\{0,1\right\}}^{{\mathrm{len}}_{\mathrm{ss}}}$**Procedure:**Choose a uniformly random key: $\mu \leftarrow_{\$}U\left({\left\{0,1\right\}}^{{\mathrm{len}}_{\mu }}\right)$Compute $\mathrm{pkh}\leftarrow \mathrm{SHAKE}(pk,{\mathrm{len}}_{\mathrm{phk}})$Generate pseudo-random values ${\mathrm{seed}}_{\mathrm{SE}}||k\leftarrow \mathrm{SHAKE}(\mathrm{phk}||\mu ,{\mathrm{len}}_{{\mathrm{seed}}_{\mathrm{SE}}}+{\mathrm{len}}_{k})$Generate pseudo-random bit string: $\left(r^{\left(0\right)},r^{\left(1\right)},\ldots ,r^{\left(2\bar{m}n+\bar{m}\bar{n}-1\right)}\right)\leftarrow \mathrm{SHAKE}(0x96||{\mathrm{seed}}_{\mathrm{SE}},\left(2\bar{m}n+\bar{m}\bar{n}\right)\cdot {\mathrm{len}}_{\chi })$Sample error matrix $S^{\prime }\leftarrow \mathrm{SampleMatrix}\left(\left(r^{\left(0\right)},r^{\left(1\right)},\ldots ,r^{\left(\bar{m}n-1\right)}\right),\bar{m},n,T_{\chi }\right)$Sample error matrix $E^{\prime }\leftarrow \mathrm{SampleMatrix}\left(\left(r^{\left(\bar{m}n\right)},r^{\left(\bar{m}n+1\right)},\ldots ,r^{\left(2\bar{m}n-1\right)}\right),\bar{m},n,T_{\chi }\right)$Generate $A\leftarrow \mathrm{Gen}({\mathrm{seed}}_{A})$Compute $B^{\prime }\leftarrow S^{\prime }A+E^{\prime }$Compute $c_{1}\leftarrow \mathrm{Pack}\left(B^{\prime }\right)$Sample error matrix $E^{\prime \prime }$: $E^{\prime \prime }\leftarrow \mathrm{SampleMatrix}\left(\left(r^{\left(2\bar{m}n\right)},r^{\left(2\bar{m}n+1\right)},\ldots ,r^{\left(2\bar{m}n+\bar{m}\bar{n}-1\right)}\right),\bar{m},\bar{n},T_{\chi }\right)$Compute $B\leftarrow \mathrm{Unpack}(b,n,\bar{n})$Compute $V\leftarrow S^{\prime }B+E^{\prime \prime }$Compute C←V+Encode(μ)Compute $c_{2}\leftarrow \mathrm{Pack}\left(C\right)$Compute $\mathrm{ss}\leftarrow \mathrm{SHAKE}(c_{1}||c_{2}||k,{\mathrm{len}}_{\mathrm{ss}})$**Return:**Ciphertext $c_{1}\left|\right|c_{2}$Shared secret ss

In the encapsulation (Algorithm 2), three noise matrices are generated: $S^{\prime }$, $E^{\prime }$, and $E^{\prime \prime }$. To create these matrices, a pseudo-random bit string is sampled according to χ. The input of the algorithm, bit strings ${\mathrm{seed}}_{A}$ and b, are used to retrieve matrices A and B. Later, they are used to calculate $B^{\prime }\leftarrow S^{\prime }A+E^{\prime }$ and $V\leftarrow S^{\prime }B+E^{\prime \prime }$. Using the matrix V added by the encoded μ (previously uniformly random generated), the matrix C is created. Then, matrices $B^{\prime }$ and C are packed, generating bit strings $c_{1}$ and $c_{2}$, which concatenated form the ciphertext. Finally, bit strings $c_{1}$, $c_{2}$, and k (pseudo-randomly generated using hash function) are hashed, creating the shared secret ss.

The decapsulation (Algorithm 3) aims to check if the ciphertext ($c_{1}\|c_{2}$) is valid. To keep it short, bit strings $c_{1}$ and $c_{2}$ are unpacked, retrieving matrices $B^{\prime }$ and C. Then, $M\leftarrow C+B^{\prime }S$ is calculated and then decoded, getting $\mu^{\prime }$. The encapsulation steps are redone, although this time generating matrices $B^{\prime \prime }$ and $C^{\prime }$. If matrices $B^{\prime \prime }$ and $C^{\prime }$ matches with matrices $B^{\prime }$ and C, the shared secret returned is the hash of $c_{1}$, $c_{2}$, and $k^{\prime }$ (pseudo-randomly generated using hash function based on $\mu^{\prime }$). Otherwise, the shared secret returned is the hash of $c_{1}$, $c_{2}$, and s (part of the secret key $sk^{\prime }$).

More information and details about the parameters, the error sampling procedure, and the lattice structure can be consulted in the official specification of FrodoKEM [26,45].
**Algorithm 3:** Decapsulation**Input:**Ciphertext: $c_{1}\left|\right|c_{2}\in {\left\{0,1\right\}}^{(\bar{m}n+\bar{m}\bar{n})D}$Secret key: $sk^{\prime }=\left(s|\right|{\mathrm{seed}}_{A}||b,S^{T}{,\mathrm{pkh})\in \left\{0,1\right\}}^{{\mathrm{len}}_{s}+{\mathrm{len}}_{{\mathrm{seed}}_{A}}+D\cdot n\bar{n}}\times Z_{q}^{n\times \bar{n}}\times {\left\{0,1\right\}}^{{\mathrm{len}}_{\mathrm{pkh}}}$**Output:**Shared secret $\mathrm{ss}\in {\left\{0,1\right\}}^{{\mathrm{len}}_{\mathrm{ss}}}$**Procedure:**Compute $B^{\prime }\leftarrow \mathrm{Unpack}\left(c_{1},\bar{m},n\right)$Compute $C\leftarrow \mathrm{Unpack}(c_{2},\bar{m},\bar{n})$Compute $M\leftarrow C+B^{\prime }S$Compute $\mu^{\prime }\leftarrow \mathrm{Decode}\left(M\right)$Parse $pk\leftarrow {\mathrm{seed}}_{A}\left|\right|b$Generate pseudo-random values: ${\mathrm{seed}}_{\mathrm{SE}}^{\prime }\left|\right|k^{\prime }\leftarrow \mathrm{SHAKE}(\mathrm{phk}||\mu^{\prime },{\mathrm{len}}_{{\mathrm{seed}}_{\mathrm{SE}}}+{\mathrm{len}}_{k})$Generate pseudo-random bit string: $\left(r^{\left(0\right)},r^{\left(1\right)},\ldots ,r^{\left(2\bar{m}n+\bar{m}\bar{n}-1\right)}\right)\leftarrow \mathrm{SHAKE}(0x96||{\mathrm{seed}}_{\mathrm{SE}}^{\prime },2\bar{m}n+\bar{m}\bar{n}\cdot {\mathrm{len}}_{\chi })$Sample error matrix $S^{\prime }\leftarrow \mathrm{SampleMatrix}\left(\left(r^{\left(0\right)},r^{\left(1\right)},\ldots ,r^{\left(\bar{m}n-1\right)}\right),\bar{m},n,T_{\chi }\right)$Sample error matrix $E^{\prime }\leftarrow \mathrm{SampleMatrix}\left(\left(r^{\left(\bar{m}n\right)},r^{\left(\bar{m}n+1\right)},\ldots ,r^{\left(2\bar{m}n-1\right)}\right),\bar{m},n,T_{\chi }\right)$Generate $A\leftarrow \mathrm{Gen}({\mathrm{seed}}_{A})$Compute $B^{\prime \prime }\leftarrow S^{\prime }A+E^{\prime }$Sample error matrix $E^{\prime \prime }\leftarrow \mathrm{SampleMatrix}\left(\left(r^{\left(2\bar{m}n\right)},r^{\left(2\bar{m}n+1\right)},\ldots ,r^{\left(2\bar{m}n+\bar{m}\bar{n}-1\right)}\right),\bar{m},\bar{n},T_{\chi }\right)$Compute $B\leftarrow \mathrm{Unpack}(b,n,\bar{n})$Compute $V\leftarrow S^{\prime }B+E^{\prime \prime }$Compute $C^{\prime }\leftarrow V+\mathrm{Encode}\left(\mu^{\prime }\right)$**if**$B^{\prime }||C=B^{\prime \prime }\left|\right|C^{\prime }$**then**($\;\;\;\;|\;\bar{k}\leftarrow k^{\prime }$**else**(($\;\;\;\;|\;\bar{k}\leftarrow s$**end**(**Return:**Shared secret $\mathrm{ss}\leftarrow \mathrm{SHAKE}(c_{1}||c_{2}||\bar{k},{\mathrm{len}}_{\mathrm{ss}})$

## 4. Implementation

Motivated by the performance results using an FPGA device for hybrid transceiver [46], this section details the proposed hardware/software implementation of the FrodoKEM scheme in an SoC device. Furthermore, it takes advantage of SoC device’s feasibility for implementing and running fast complex algorithms [47], such as the FrodoKEM-640 variant using the SHAKE128 hash function. This variant matches or exceeds the brute-force security of AES-128.

Regarding practical applications, the literature points out the following approaches to implementing PQC schemes [48]:**Software-based:** the implementation is carried out in microprocessors, microcontrollers, and ARM-processors. In other words, it is totally in software. The implementation is concluded in a short period, and the cost is low; however, it may not offer real-time performance.**Hardware/software-based with soft-core processors:** it uses an FPGA with a soft-core processor implemented inside it, typically MicroBlaze or Nios II. Usually, the processor holds the main application, and the FPGA acts as a hardware accelerator. The implementation takes more time and costs than the software implementation; however, it offers real-time performance.**Hardware/software-based with hard-core processors:** it is similar to the previous approach. The difference is that a hard-core processor, typically an ARM, is integrated into an SoC device, in which an FPGA device exists. The hard-core processor does not consume any FPGA area. Moreover, the implementation takes more time and costs than the software implementation; however, it offers better real-time performance than the previous approach.**Hardware-based:** it refers to the hardware implementation in an FPGA device. The implementation takes much more time and costs than the previous approaches; however, it offers the best real-time performance.

It is worth mentioning that SM equipment is supposed to be a commodity in the electricity sector, which requires low production costs and capacity to embed a PQC scheme. Therefore, full software-based implementation may not be fast enough to support a PQC scheme and comply with the constraints of SM systems. On the other hand, a complete hardware-based implementation would be expensive. In this regard, a hardware/software-based implementation is the alternative with the most attractive trade-offs.

To the author’s knowledge, some implementations of PQC schemes using the first and fourth approaches [18,49,50,51], but there is a gap in the literature concerning the second and third approaches. Regarding the third approach, only high-speed implementations are found in the literature [48], rather than the lightweight implementation targeted by this paper. The second and third approaches have a high potential to present a considerably better performance than software-based implementation. Also, they can offer good performance compared to hardware-based implementation, even considering the overhead of exchanging data between ARM and FPGA components. Regarding the second and third approaches, the hardware/software-based with a hard-core processor has a higher potential because the processor is hardware-implemented. In this sense, the third approach, hardware/software-based with a hard-core processor, seems to be the best approach to implement a PQC scheme in a hardware-constrained device, such as a SM equipment.

### 4.1. Preliminary Analysis

To identify the main bottlenecks of a benchmark implementation of FrodoKEM, we executed the code from [39] on a Cortex-A9 ARM processor. After a detailed analysis, three operations stood out as the most time-consuming. As expected, the operation B←AS+E, used in the key pair generation process, has a high computational burden because of its large matrix-by-matrix multiplication, which requires long loops with addition and multiplication operations. For the same reason, the operation $B^{\prime }\leftarrow S^{\prime }A+E^{\prime }$ is even more expensive, because it is used in both encryption and decryption algorithms. Finally, the most expensive operation is the SHAKE128 hash function, a specific version of the SHAKE hash function. It is time-consuming in all three algorithms due to its loop-based structure. As mentioned above, all operations demand relevant computational burdens because of the large size of FrodoKEM keys and the use of a public or private key, directly or indirectly. Table 1 summarizes the relative execution time of FrodoKEM-640 in the aforementioned ARM processor.

With this preliminary analysis accomplished, our strategy for accelerating the code execution is to implement the three operations above in hardware and keep the software part (including the three operations) as close as possible to the original to ensure fairness in further comparisons.

### 4.2. Main Components

The MicroZed 7010 board was chosen to implement the two matrix-by-matrix multiplications and the SHAKE128 hash function in an FPGA and the rest of the codes in an ARM processor. The MicroZed board is a low-cost System on Module (SoM) based on the Xilinx Zyqn-7000 SoC. In addition to the Zyqn-7000, the module contains 1 GB of DDR3 SDRAM, 128 Mb of QSPI Flash, a 33.33 MHz oscillator, and other functions and interfaces.

The Xilinx Zyqn-7000 SoC used is a XC7Z010-1CLG400C, which has a Processing System (PS) and Programmable Logic (PL). The PS is based on an ARM Cortex-A9 with ARMv7-architecture. On the other hand, the PL is based on an FPGA with 28K Programmable Logic Cells, 17,600 Lookup Tables (LUTs), 35,200 Registers, 60 Block Random Access Memorys (BRAMs) with 36 Kb each, and 80 Digital Signal Processing (DSP) blocks [25]. Using a Phase-Locked Loop (PLL), 666.66 MHz and 100 MHz clocks are derived from the 33.33 MHz built-in oscillator to feed PS and PL, respectively.

### 4.3. Structure

The schematic diagram representing the operation of the FrodoKEM scheme is shown in Figure 2. It can be divided into three main instances: PS, Interconnect, and PL.

The PS, based on an ARM Cortex-A9, is where the benchmark implementation is placed. The interconnect instance is responsible for providing an interface between the PL using the AXI-MM protocol. Finally, the PL, based on an FPGA, is where the operations AS←A×S and $S^{\prime }A\leftarrow S^{\prime }\times A$ together with the SHAKE128 hash function are hardware implemented.

### 4.4. Benchmark Implementation

The benchmark implementation refers to the software implementation of the FrodoKEM scheme in the ARM processor. In this sense, the code was transferred entirely and adapted for the ARM processor, ensuring it complies with the software implementation detailed in [39]. The software implementation can be divided into three main parts that individually correspond to Algorithms 1–3.

Moreover, it is worth detailing the processing of the operations B←AS+E and $B^{\prime }\leftarrow S^{\prime }A+E^{\prime }$, and SHAKE128 hash function. As mentioned earlier, the matrix A is large, and consequently, it is unfeasible to generate it all at once in an embedded device due to its memory constraints. Therefore, matrix A is generated slightly differently as presented in Algorithm 1.

To reduce the necessity of large memory, it is proposed to generate parts of the matrix A on-the-fly and overwritten by a new part after use. Due to this technique, the FrodoKEM scheme can be embedded in a hardware-constrained device. Therefore, the matrix-by-matrix multiplication AS has three main loops. Four rows of the matrix A are generated on-the-fly using the SHAKE128 hash function in the outer loop. The middle loop is responsible for selecting one column from the matrix S, previously fully generated. In the inner loop, four elements, one element of each of four generated rows of the matrix A, are multiplied by one element from the selected column of the matrix S, and the four results are accumulated. Algorithm 4 shows in detail the process, which is also illustrated in Figure 3.   
**Algorithm 4:**B←AS+E**Input:**Matrix S: $S\in Z_{q}^{n\times \bar{n}}$Matrix E: $E\in Z_{q}^{n\times \bar{n}}$Seed: ${\mathrm{seed}}_{A}\in {\left\{0,1\right\}}^{{\mathrm{len}}_{{\mathrm{seed}}_{A}}}$**Output:**Matrix B: $B\in Z_{q}^{n\times \bar{n}}$**Procedure:**$A\in Z_{q}^{4\times n}\leftarrow {\left\{0\right\}}^{4\times n}$B←E**for**i←0,4,8,…,n **do** $A\left[0..3\right]\leftarrow shake128\left({\mathrm{seed}}_{A}\right)$ **for**
$k\leftarrow 0,\ldots ,\bar{n}$ **do**  $sum\in Z_{q}^{4}\leftarrow {\left\{0\right\}}^{4}$  **for** j←0,…,n **do**   sum[0..3]←sum[0..3]+A[0..3,j]·S[k,j]  **end**  B[i+0..3,k]←B[i+0..3,k]+sum[0..3] **end****end**

The operation $B^{\prime }\leftarrow S^{\prime }A+E^{\prime }$ occurs differently from the previous one. It also (re)generates the matrix A on-the-fly, although the multiplication process must be adapted. It occurs because the regenerated matrix A, used in the encapsulation and decapsulation processes, must be the same as used in the key pair generation. Therefore, each SHAKE128 hash function will generate a row of the matrix A which will slightly complicate the logic of the multiplication process, considering that the matrix A in this operation is on the right-hand side and it is no longer possible to multiple an entire row of the matrix A by an entire column of the matrix S, in this case, the matrix $S^{\prime }$. The Algorithm 5 presents this matrix-by-matrix multiplication process, which is also pictured in Figure 4.
**Algorithm 5:**$B^{\prime }\leftarrow S^{\prime }A+E^{\prime }$**Input:**Matrix $S^{\prime }$: $S^{\prime }\in Z_{q}^{n\times \bar{n}}$Matrix $E^{\prime }$: $E^{\prime }\in Z_{q}^{n\times \bar{n}}$Seed: ${\mathrm{seed}}_{A}\in {\left\{0,1\right\}}^{{\mathrm{len}}_{{\mathrm{seed}}_{A}}}$**Output:**Matrix $B^{\prime }$: $B^{\prime }\in Z_{q}^{n\times \bar{n}}$**Procedure:**$A\in Z_{q}^{4\times n}\leftarrow {\left\{0\right\}}^{4\times n}$$B^{\prime }\leftarrow E^{\prime }$**for**i←0,4,8,…,n **do** $A\left[0..3\right]\leftarrow shake128\left({\mathrm{seed}}_{A}\right)$ **for**
$k\leftarrow 0,\ldots ,\bar{n}$ **do**  $sum\in Z_{q}^{n}\leftarrow {\left\{0\right\}}^{n}$  **for** j←0,…,4 **do**   **for** p←0,…,n **do**    $sum\left[p\right]\leftarrow sum\left[p\right]+S^{\prime }\left[k,i+j\right]\cdot A\left[j,p\right]$   **end**  **end**  **for** j←0,…,n **do**   B[k,j]←B[k,j]+sum[j]  **end** **end****end**

Finally, SHAKE128 [52] is a hash function with an output length of 256-bits and a security level of 128-bits. SHAKE128 is an instance of SHA-3, the latest member of the Secure Hash Algorithm family of standards, released by NIST. The SHA-3 family is based on the sponge construction [53], which is shown in Figure 5. The SHAKE128 hash function uses the Keccak-f[1600] function as the transform function composed of five permutation steps, whose parameters are the block size *r* equal to 1344 bits and its capacity *c* equal to 256 bits, resulting in the internal state with 1600 bits. The SHAKE128 hash function can be divided into two main parts: the absorbing and squeezing phases. In the absorbing phase, the input blocks (message) are XORed into the bit string *r* of the internal state. Then, the internal state is inputted in the Keccak-f(1600) function. When the entire input message is absorbed, the squeeze phase begins. In this case, the outputted blocks are read from the bit string *r* of the state, alternating with the Keccak-f(1600) function until the desired output size is reached.

### 4.5. The Proposed Hardware/Software Implementation

Based on the results presented in Table 1, the most time-consuming operations are implemented in hardware (i.e., the FPGA) to accelerate the execution of the FrodoKEM scheme. These implementations are detailed in the following subsections.

#### 4.5.1. The Operation AS←A×S

The operation B←AS+E, which consumes more than 8% of the total time, has its matrix-by-matrix multiplication implemented in hardware. The operation of adding matrix E is performed in software, as it is not an expensive operation and would not bring significant time saving, under the assumption that the transferring time of the matrix E from PS to PL is taken into account.

Figure 6 shows the hardware schematic of the operation AS←A×S. The schematic consists of four main instances: BRAMs of the matrix A, BRAMs of the matrix S, a multiplier instance, and BRAMs of the matrix AS. Note that BRAMs are Block Random Access Memory where a larger amount of data are stored.

As mentioned earlier, matrix A is generated on-the-fly to save memory resources. Four rows are generated and transmitted from the PS to the BRAMs of the matrix A, located in PL, to be stored. Four 320×32-bits BRAMs are ready to receive these rows. Each generated row is composed of 640×16-bits. To speed up the data transfer process, 32-bits are transferred at a time, which means that two subsequent values are concatenated and stored in the BRAMs. The matrix S is transmitted from PS to PL. The whole matrix is transferred since it has already been fully generated in PS. The transmission process follows the same 32-bits transmit principle as with matrix A, although, for the matrix S a 2560×32-bits BRAM is used.

The multiplication process begins when both matrices are completely stored in their BRAMs. From BRAMs of the matrix A the *n*-element of each BRAM is read. As 2×16-bits words are stored concatenated in a 32-bits BRAM position, eight values are loaded from memory and sent to the multiplier instance. In parallel, the *n*-element of the matrix S is also read. For the same reason as the concatenated storage, 2×16-bits values are loaded and transferred to the multiplier instance.

In the multiplication instance, each 32-bits element is split into 2×16-bits words using the function *S*. Then, the multiplications are properly performed, and their results are added to the previous iteration results. Any concern about overflow issues is needed, which is an advantage of the FrodoKEM scheme in saving hardware resources. While the multiplication is performed, the following values from the BRAMs are loaded, keeping the pipeline full to achieve maximum performance. When all elements of the BRAMs are loaded and processed, the 4×16-bits accumulated values are transferred to the BRAMs of the matrix AS, which has four 1280×16-bits BRAMs.

The next four rows of the matrix A must be generated and transferred to PL. The matrix S has already been completely transferred in the first iteration and should not be sent again. The next iteration begins, and the process mentioned above is performed again. When all iterations are complete, the BRAMs of the matrix AS store the AS←A×S result and transfers it to PS using a 32-bits bus.

#### 4.5.2. The Operation $S^{\prime }A\leftarrow S^{\prime }\times A$

This is responsible for more than 30% of the execution time. The operation $B^{\prime }\leftarrow S^{\prime }A+E^{\prime }$ also has its matrix-by-matrix multiplication implemented in hardware. Similarly to the operation B←AS+E, the addition operation of the matrix $E^{\prime }$ is carried out in software due to the same reasons.

In Figure 7 we can see the schematic representation of the operation $S^{\prime }A\leftarrow S^{\prime }\times A$. It also has four instances: BRAMs of the matrix $S^{\prime }$, BRAMs of the matrix A, a multiplier instance, and BRAM of the matrix $S^{\prime }A$.

Two 640×32-bits BRAMs are used to store half of the matrix $S^{\prime }$. As mentioned earlier, concatenated values are transferred from PS to PL using 32-bits bus; therefore, each BRAM can be split in half (upper 16-bits and lower 16-bits), storing two concatenated word by word columns of the matrix $S^{\prime }$, giving a total of four columns. On the other hand, the matrix A and its transfer process have exactly the same configuration as used in the operation AS←A×S: four 320×32-bits BRAMs and 32-bits bus for transfer.

After the matrices are received, multiplication is performed. The *n*-element of each matrix $S^{\prime }$ BRAM are loaded. Since 2×16-bits words are stored concatenated in a 32-bits BRAM position, four values are read and sent to the multiplier instance in 16-bits buses. In parallel, the *p*-element of each BRAM of the matrix A are read. Only the upper or lower 16-bits of the four loaded values are sent to the multiplier instance, depending on the *p* parity.

Therefore, the multiplier instance receives 8×16-bits values, four from each matrix. Finally, the values are correctly multiplied, and the results are added, resulting in a 16-bits value. Immediately, the multiplication and addition process results are sent to matrix $S^{\prime }A$, stored in one 5120×16-bits BRAM, to be added to a previous value stored in a particular position. Simultaneously, new values from matrices $S^{\prime }$ and A are loaded to keep the pipeline full, restarting the process.

When all iterations are completed, the other half of the matrix $S^{\prime }$ and new rows of the matrix A must be transferred to PL. When the process ends, the BRAMs of the matrix $S^{\prime }A$ will store the result of operation $S^{\prime }A\leftarrow S^{\prime }\times A$ and it is transmitted to PS using a 32-bits bus.

#### 4.5.3. SHAKE128 Hash Function

FrodoKEM’s most time-consuming operation is the SHAKE128 hash function, which is responsible for almost 54% of the total execution time.

The proposed scheme can be organized into three main instances: a BRAM, the SHAKE128 hash function instance, and a Keccak-f[1600], as illustrated in Figure 8.

The BRAM is a 2583×64-bits memory. This size was chosen based on the maximum size that FrodoKEM needs. This BRAM is responsible for storing the input values (message) received by the 32-bits bus. Each 32-bits word is a concatenation of 4×8-bits characters. 2×32-bits words or 8×8-bits characters are concatenated and stored in the BRAM.

When all values are received and stored in the BRAM, 168-bytes are sent to the SHAKE128 hash function instance via a 64-bits bus, starting the absorb phase. These bytes built the internal state. If necessary in the absorb phase, the Keccak-f[1600] function is called, and the entire internal state is sent to the Keccak-f[1600] function, which performs its five steps (θ, ρ, π, χ, and ι) in a single clock. Then, the new scrambled internal state returns to the SHAKE128 hash function. Next, another 168-bytes are loaded from BRAM and the process is repeated until the entire inputted message is read, finalizing the absorb phase.

When the absorb phase ends, the squeeze phase starts using the Keccak-f[1600] function. To save resources, the first 168-bytes of the internal state of each step in the squeeze phase are stored in the same BRAM, which previously stored the inputted message and now stores the output values (cipher). When the desired output length is reached, the process is complete, and the BRAM uses the 32-bits bus to send the stored output values back to PS.

## 5. Performance Evaluation

The FPGA hardware resource usage and execution times of FrodoKEM are presented in this section. First, Section 5.1 details the hardware resources of the FPGA component of the SoC device that is used by the proposed hardware/software implementation. In the sequel, Section 5.2 focuses on the time consumption analysis of the operations AS←A×S and $S^{\prime }A\leftarrow S^{\prime }\times A$, and the SHAKE128 hash function, for both benchmark and proposed hardware/software implementations.

### 5.1. Hardware Resource Usage Analysis

The hardware resource usages of operations AS←A×S and $S^{\prime }A\leftarrow S^{\prime }\times A$, and the SHAKE128 hash function are listed in Table 2.

Table 2 shows that the operations AS←A×S and $S^{\prime }A\leftarrow S^{\prime }\times A$ use very few Slice LUTs and Slice Registers in comparison to the SHAKE128 hash function. It relies on the logic circuit complexity necessary to implement the matrix-by-matrix multiplications, which is much simpler than the one required to implement the SHAKE128 hash function.

The operation AS←A×S uses 10 BRAMs, while the operation $S^{\prime }A\leftarrow S^{\prime }\times A$ uses only 6 BRAMs. This difference occurs because the former stores the entire matrix S in the PL, while the latter stores only half of it. The SHAKE hash function uses 7.5 BRAMs, enough memory space to store the larger cipher required by FrodoKEM.

Finally, the operation AS←A×S uses 8 DSP blocks, which is in accordance with Figure 6. Contrastingly, the operation $S^{\prime }A\leftarrow S^{\prime }\times A$ uses only 4 DSP blocks, as presented in Figure 7. The SHAKE128 hash function does not use any DSP block.

### 5.2. Time Consumption Analysis

Three different analyses are considered to compare the implementation of the FrodoKEM scheme with and without hardware acceleration: (i) comparison between different levels of hardware implementations, (ii) most time-consuming operations, and (iii) hardware-only analysis. In this section, we evaluate the implementations using execution time and processing time parameters. The former considers a routine’s processing time plus the time for exchanging data, while the latter refers to the time to compute the routine only, which excludes the time for exchanging data. The measured times were acquired using a dedicated timer implemented in the PL.

#### 5.2.1. Comparison between Different Levels of Hardware Implementations

Now, we compare different implementations of FrodoKEM that ranges from the benchmark implementation up to the proposed hardware/software implementation. To do so, we present all combinations of software and hardware implementations of AS←A×S and $S^{\prime }A\leftarrow S^{\prime }\times A$, and SHAKE128 hash function because these operations are the most time consuming in the software implementation, see Table 1. The results attained by different levels of software and hardware implementations, which range from the benchmark implementation up to the proposed hardware/software implementation, are listed in Table 3. Note that the key pair generation, encapsulation, and decapsulation algorithms were executed eight times each, covering all possible hardware and software implementation combinations of these operations. The first three columns refer to the choice of type implementation for each operation, and the following four columns present the execution time of each algorithm and the total time demanded by all of them. Moreover, the last column presents the relative time improvement ($\alpha_{\mathrm{TI}}$), in percentage, comparing the total execution time of the benchmark implementation (first row) with the total execution time of different levels of the hardware/software implementations, which is given by
(1)$$\alpha_{\mathrm{TI}}=1-\left(T_{\mathrm{TETH}}/T_{\mathrm{TETB}}\right),$$
in which $T_{\mathrm{TETH}}$ is the total execution time from the second row onwards and $T_{\mathrm{TETB}}$ is the total execution time of the benchmark implementation (first row) in Table 3. In the following, we discuss the results presented in a few rows because the analysis of the others can be easily figured out from our discussions.

The second row in Table 3 details a partial hardware implementation, in which the operation AS←A×S is hardware-implemented. In this case, an improvement of 3.60% compared to the benchmark implementation is obtained. This low improvement is because such operation is simple to implement and it is performed only once in the key pair generation algorithm.

The third row in Table 3 lists another partial hardware implementation, in which the operation $S^{\prime }A\leftarrow S^{\prime }\times A$ is hardware-implemented. It reached a 20.34% of improvement compared to the benchmark implementation. This considerable improvement is because the software implementation of $S^{\prime }A\leftarrow S^{\prime }\times A$ is more complex than the software implementation of AS←A×S, consequently, a parallelizable hardware implementation brings significant advantages.

Now, paying attention to the fifth row in Table 3, we can see that the hardware implementation of the SHAKE128 hash function brings an improvement of 44.20%. This improvement can be explained due to a fast and straightforward implementation in hardware compared with a costly loop-based software implementation. Moreover, SHAKE128 hash function is often used in all three algorithms, increasing its impact on execution time.

The analysis of the last row in Table 3 shows that the hardware implementation of these three operations provides a relative time improvement of 68.15%, which comprise the summation of individual time improvement. Therefore, when hardware acceleration is used, the three operations can be performed with less than one-third of the time required by the benchmark implementation (i.e., the software implementation), dropping the execution time from more than 1700 ms to less than 560 ms.

Last but not least, better results in terms of hardware acceleration can be achieved if the data transmission is sped-up. For instance, using buses larger than 32-bits between the DSP and FPGA devices and more DSP blocks for improving the parallel data processing (e.g., matrix-by-matrix multiplication).

#### 5.2.2. Most Time-Consuming Routines

Now, we intend to individually analyze the operations B←AS+E and $B^{\prime }\leftarrow S^{\prime }A+E^{\prime }$, and the SHAKE128 hash function (i.e., without considering its impact on the whole implementation). To carry out this analysis, we assume that all routines necessary to generate matrices B and $B^{\prime }$ are considered, not just the matrix-by-matrix multiplication. Table 4 lists the attained results related to these three operations, presenting the execution time and average execution time in software and hardware, as also the relative time improvement ($\beta_{\mathrm{TI}}$), which is given by
(2)$$\beta_{\mathrm{TI}}=1-\left(T_{\mathrm{ETH}}/T_{\mathrm{ETS}}\right),$$
where $T_{\mathrm{ETH}}$ is the execution time in hardware and $T_{\mathrm{ETS}}$ the execution time software.

The hardware implementation of operations B←AS+E and $B^{\prime }\leftarrow S^{\prime }A+E^{\prime }$ achieve satisfactory results. The numbers of iterations of these operations are one and two, respectively, in the FrodoKEM scheme. The former operation spends 122.08 ms and 59.14 ms when software- and hardware-implemented, demanding average iteration times of 122.08 ms and 59.14 ms, respectively, and achieving an improvement of 51.56%. In contrast, the latter operation spends 618.65 ms and 224.78 ms when software- and hardware-implemented. Also, it attains average iteration times of 309.32 ms and 112.39 ms, respectively, and an improvement of 63.66%.

Moreover, the number of iterations of the SHAKE128 hash function is 1930 in one execution of the FrodoKEM scheme. As a result, the execution time for each iteration for the software and hardware implementations are 0.526 ms and 0.121 ms, respectively. Consequently, the hardware implementation reaches a relative time improvement of 76.90%.

#### 5.2.3. Hardware-Only Time Analysis

Another meaningful analysis is the hardware-only time analysis. This analysis focused on comparing each routine’s execution time and processing time, both in hardware. To carry out this analysis, in the operations B←AS+E and $B^{\prime }\leftarrow S^{\prime }A+E^{\prime }$, only the matrix-by-matrix multiplication is considered because it is the part performed in hardware. This analysis is numerically evaluated by using the so-called relative time ($\gamma_{T}$) parameter, which is given by
(3)$$\gamma_{T}=\left(T_{\mathrm{APT}}/T_{\mathrm{AET}}\right),$$
where $T_{\mathrm{APT}}$ is the average processing time in hardware per iteration and $T_{\mathrm{AET}}$ is the average execution time in hardware per iteration. The attained results are presented in Table 5.

As can be seen, the processing time in hardware of operations AS←A×S and $S^{\prime }A\leftarrow S^{\prime }\times A$, and the SHAKE128 hash function are much shorter than he execution time in hardware, representing 6.98%, 13.06%, and 4.07%, respectively. These results mean that most of the time is spent receiving data from PS or sending data back to PSwhile only a short time is dedicated to performing the operations in hardware.

## 6. Conclusions

This work has investigated the feasibility of implementing a PQC scheme in a hardware-constrained SM equipment for ensuring the security of SM data traveling through data networks. In this context, FrodoKEM, a post-quantum lattice-based scheme, was detailed and, in the sequel, implemented in an SoC device.

Numerical results have shown that the proposed hardware/software implementation of the most time-consuming and complex routines of FrodoKEM in hardware results in a one-third reduction of the execution time compared to the benchmark implementation (i.e., software implementation). According to the execution time analysis, three operations consume almost all execution time. The separate implementation of these three operations in hardware reaches an improvement of 68.15% (i.e., the execution time reduces from 1.7 s to less than 0.6 s). Consequently, these results show that it is possible to implement the FrodoKEM scheme in hardware-constrained equipment, such as SM, in which an SoC device is used. In such a case, the FPGA component is dedicated to running the most time-consuming routines.

Overall, it has shown that implementing the FrodoKEM scheme is feasible to secure SMs data traveling through data networks. Last but not least, all discussions and analyses constitute baselines to adopt PQC schemes in SM systems.

## 7. Future Work

Other analyses regarding the FrodoKEM scheme and our proposed hardware/software implementation, such as side-channel attack analysis and its impacts on security, were beyond the scope of this work and, consequently, have been left for future work. Furthermore, it would be of the utmost importance to compare the proposed hardware/software implementation of the FrodoKEM scheme with the hardware/software implementations of other candidates of the NIST post-quantum standardization process. Finally, it would be interesting to analyze the FrodoKEM scheme in a non-controlled environment as a key encapsulate mechanism embedded, such as in a data network in which SM equipment communicates with an MDMS.

## Figures and Tables

**Figure 1 sensors-22-07214-f001:**
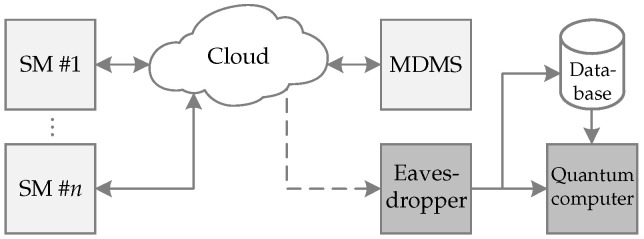
A typical security breach scenario in which an eavesdropper performs a sniffing attack and makes use of a quantum computer to decipher the overheard SM data.

**Figure 2 sensors-22-07214-f002:**
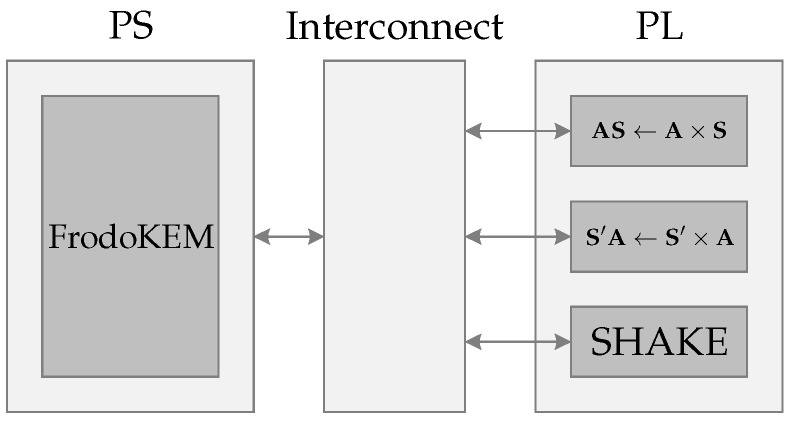
Schematic representation of the structure. The arrows represent 32-bit buses. Control signals have been omitted.

**Figure 3 sensors-22-07214-f003:**
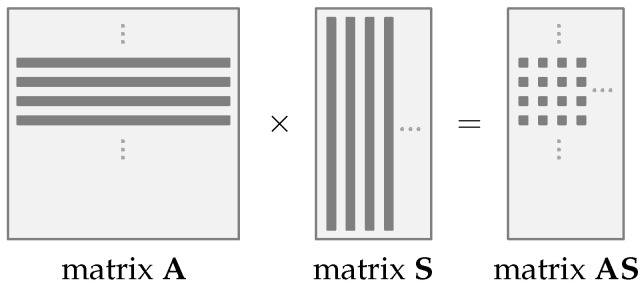
Operation AS←A×S. Multiplication of four rows of the matrix A by matrix S.

**Figure 4 sensors-22-07214-f004:**
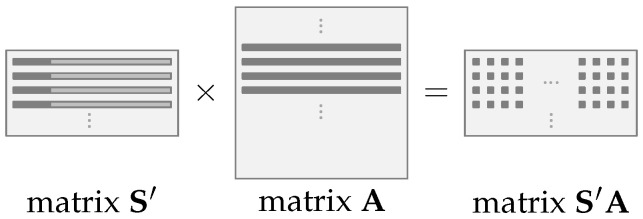
Operation $S^{\prime }A\leftarrow S^{\prime }\times A$. Multiplication of first four elements of the first four rows of the matrix $S^{\prime }$, cmatrix A.

**Figure 5 sensors-22-07214-f005:**
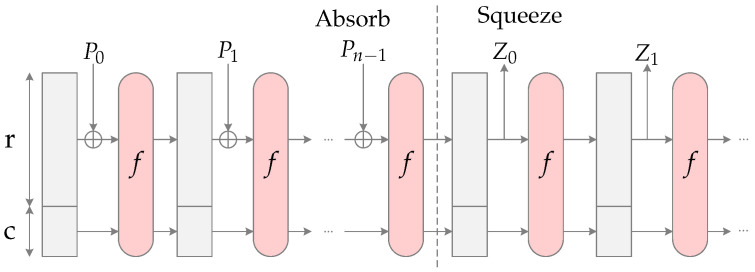
Sponge construction.

**Figure 6 sensors-22-07214-f006:**
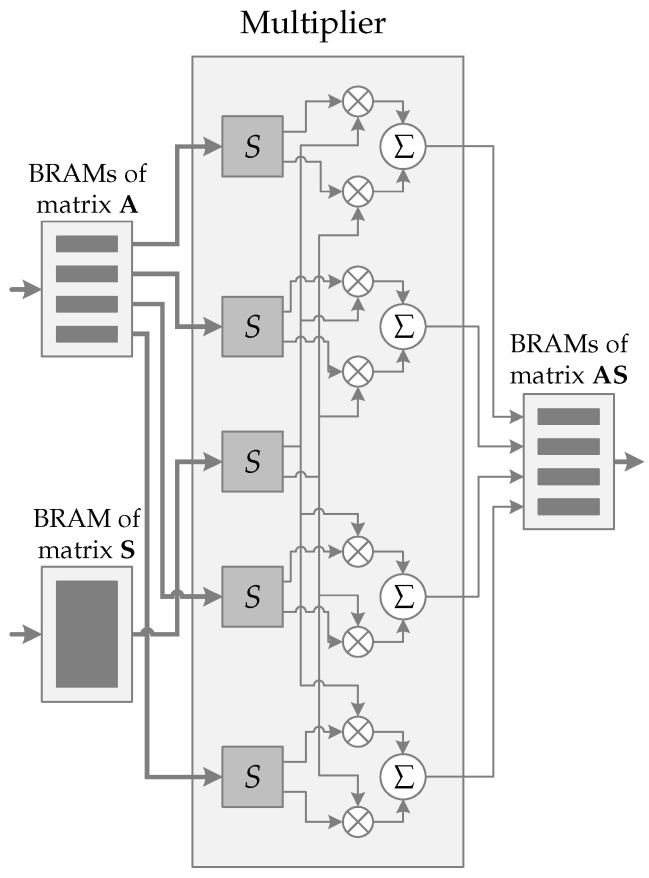
Schematic representation of AS←A×S. Thick arrows represent 32-bits buses and thin arrows 16-bits buses. Control signals have been omitted.

**Figure 7 sensors-22-07214-f007:**
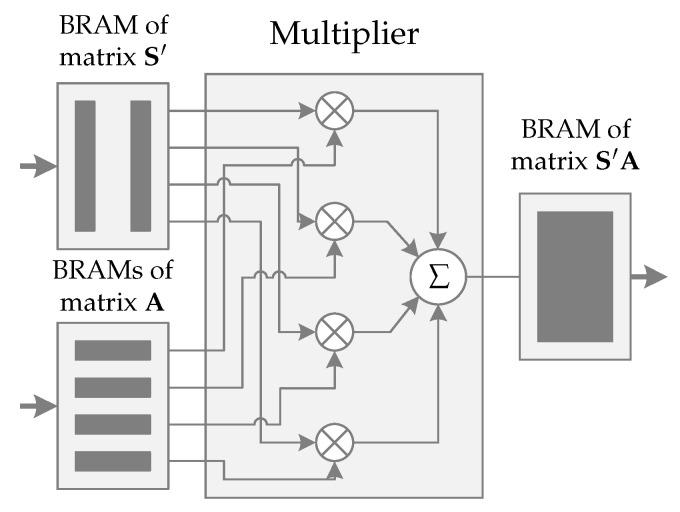
Schematic representation of $S^{\prime }A\leftarrow S^{\prime }\times A$. Thick arrows represent 32-bits buses and thin arrows 16-bits buses. Control signals have been omitted.

**Figure 8 sensors-22-07214-f008:**
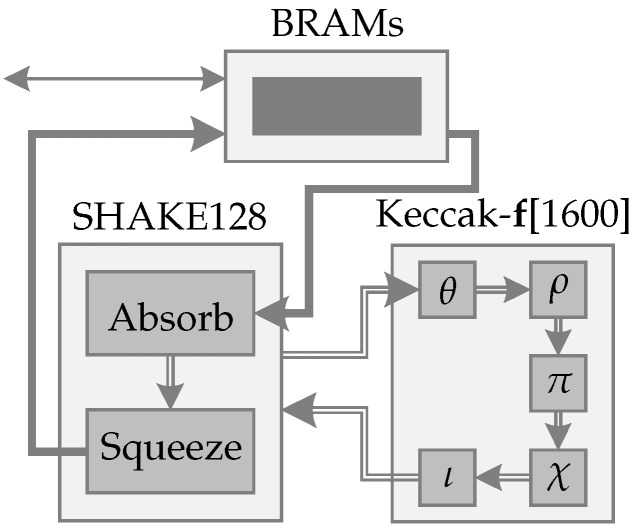
Block diagram representation of the SHAKE128 hash function. Thin line represent 32-bits buses, thick line represent 64-bits buses, and double lines represent 1600-bits buses. Control signals have been omitted.

**Table 1 sensors-22-07214-t001:** The relative execution time of the FrodoKEM-640 scheme, in percentage.

Operation	(%)
B←AS+E	8.22
$B^{\prime }\leftarrow S^{\prime }A+E^{\prime }$	30.94
SHAKE128	53.90
Others	6.94

**Table 2 sensors-22-07214-t002:** Hardware resource usage.

Operation	Slice LUTs	Slice Registers	Block RAM	DSP
AS←A×S	359	272	10	8
$S^{\prime }A\leftarrow S^{\prime }\times A$	268	301	6	4
SHAKE128	5395	3648	7.5	0

**Table 3 sensors-22-07214-t003:** The execution time for different levels of hardware implementations, in milliseconds (ms).

SHAKE128	$S^{\prime }A\leftarrow S^{\prime }\times A$	AS←A×S	Key Pair Generation	Encap- Sulation	Decap- Sulation	Total Execution Time	Relative Time Improvement ($\alpha_{\mathrm{TI}}$)
Benchmark	Benchmark	Benchmark	467.25	644.74	645.35	1757.33	-
Benchmark	Benchmark	Proposed	404.03	644.72	645.27	1694.02	3.60%
Benchmark	Proposed	Benchmark	467.33	466.01	466.52	1399.86	20.34%
Benchmark	Proposed	Proposed	404.09	466.02	466.62	1336.73	23.93%
Proposed	Benchmark	Benchmark	209.03	384.18	387.31	980.52	44.20%
Proposed	Benchmark	Proposed	146.29	384.18	387.31	917.77	47.77%
Proposed	Proposed	Benchmark	208.77	205.10	208.21	622.09	64.60%
Proposed	Proposed	Proposed	146.35	205.11	208.26	559.72	68.15%

**Table 4 sensors-22-07214-t004:** Execution time analysis, in ms.

Operation	Iterations	Execution Time in Software	Average Execution Time in Software per Iteration	Execution Time in Hardware	Average Execution Time in Hardware per Iteration	Relative Time Improvement ($\beta_{\mathrm{TI}}$)
B←AS+E	1	122.08	122.08	59.14	59.14	51.56%
$B^{\prime }\leftarrow S^{\prime }A+E^{\prime }$	2	618.65	309.32	224.78	112.391	63.66%
SHAKE128	1930	1015.88	0.526	234.71	0.121	76.90%

**Table 5 sensors-22-07214-t005:** Hardware-only execution and processing time, in ms.

Operation	Iterations	Execution Time in Hardware	Average Execution Time in Hardware per Iteration	Processing Time in Hardware	Average Processing Time in Hardware per Iteration	Relative Time ($\gamma_{T}$)
AS←A×S	1	58.61	58.61	4.09	4.094	6.98%
$S^{\prime }A\leftarrow S^{\prime }\times A$	2	125.43	62.71	16.38	8.190	13.06%
SHAKE128	1930	200.78	0.104	8.17	0.004	4.07%

## Data Availability

Not applicable.

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
