# Peer review of "A System-on-a-Chip Implementation of a Post-Quantum Cryptography Scheme for Smart Meter Data Communications"

_sensors, 2022, doi:10.3390/s22197214_

Round 1

Reviewer 1 Report

Please refer to the review report.

Author Response

1) First of all, we are delighted with your comments. Regarding the homomorphic encryption, according to the Algorithm Specifications and Supporting Documentation of the FrodoKEM [26], its parameters can be altered to support features beyond PKE and KEM. For instance, by changing the integer modules $q$ parameter, FrodoPKE can easily support homomorphic additions or multiplications by small public scalars on the ciphertext. Furthermore, if a larger moduli is chosen, it can be made into a fully homomorphic encryption (FHE) scheme. While FrodoPKE can be made FHE, FrodoKEM can not. The FrodoKEM is designed to be an IND-CCA2 scheme, which means that the ciphertext is authenticated (i.e., it is not malleable). Overall, the discussion about homomorphic encryption is beyond the scope of this work, but it is an interesting research topic for investigation. Unfortunately, the scope of our investigation was not designed to subsidize a discussion about homomorphic encryption using a PQC scheme for energy data aggregation. Consequently, we can not answer this question. 

2) First of all, we are thankful for your comments. Yes, the FrodoKEM can be implemented in a simple MCU using an implementation technique that regenerates the Public-key (PK) on-the-fly, see Section 4.4 for details. Giving some background, usually, the side with more power processing (in this case, the MDMS) is who starts the Key Encapsulation Mechanism (KEM), generating a PK and a Secret-Key (SK). The SK must be stored in the MDMS and never be shared. Also, the PK is sent to the other side (in our case, an SM equipment). However, this PK has a large size, almost 20 kBytes for FrodoKEM-640. In this sense, instead of sending the PK, which is a large block of bits, the FrodoKEM scheme sends only the seed (which is very small) required to regenerate the same PK using the SHAKE128 hash functions. When the SM receives this seed, it uses the SHAKE128 hash function to (re)generate on-the-fly the PK (piece by piece) and, later, after the required processing (see Algorithm 2), achieve a shared secret and also a ciphertext. The former will be used as a symmetric key for secure communication using symmetric cryptography, while the latter will be sent to the MDMS, which will use it to achieve the same symmetric key as the SM equipment. In this sense, the SM has only to be capable of holding a tiny piece of the PK, consuming low hardware resources. In other words, the FrodoKEM can be implemented in a simple MCU, but it will probably lack performance. Therefore, our investigation focuses on using an SoC device to perform PQC because it can overcome this problem.

3) First of all, we are thankful for your comments. We have included a paragraph in the  Introduction Section to address your concern. This paragraph provides a concise discussion about previous works and the motivations for our work. In this work, we are interested in investigating the extent to which the application of software and hardware implementations in an SoC device can benefit PQC in smart metering systems. We are currently working on the hardware implementation of another PQC scheme, and hopefully, we will show some comparisons soon. 

Reviewer 2 Report

The necessity of ensuring the security of smart meter data traveling through data networks will be a challenge in the era of quantum computing because a quantum computer might exploit characteristics of well-established cryptographic schemes to reach a successful breach of security. Post-quantum cryptography (PQC) is an important part of quantum cryptography. This manuscript has investigated the feasibility of implementing a PQC scheme in hard-ware-constrained SM devices or dedicated cryptographic modules for ensuring the security of SM data traveling through data networks. In this context, FrodoKEM, a post-quantum lattice-based scheme, was detailed and, in the sequel, implemented in an SoC device. Experimental results show that the processing time to run the FrodoKEM scheme implemented in an SoC device reduces to one-third of that obtained by the reference implementation. Moreover, the attained processing time and hardware resource usage of this SoC-based implementation of the FrodoKEM scheme shows that lattice-based cryptography may fit smart meter devices. So, it makes a useful contribution to the subject. Major revisions are needed before it is finally accepted. 

In this manuscript, the security analysis of the proposed scheme is not presented. The authors should apply the information theory, and how much information the attacker obtains in the different attacks. The authors need to show more persuasive reasons for readers that their proposal offers a better solution.

Author Response

First of all, we very pleased with your comments. We want to point out that this papers does not address the security of FrodoKEM. The best results on cryptoanalytic attacks are presented in the paper [26]. 

Our work focuses on the feasibility of implementing FrodoKEM in hardware-constrained equipment, in which an SoC device can be applied. In this sense, we detailed a discussion on how FrodoKEM can be implemented to benefit SM equipment that uses an SoC device. Due to the deepness and complexity of our investigation, only the implementation (hardware/software) aspects are addressed in this work. 

Last, we are still investigating PQC for smart metering. We also developed a setup to analyze the side-channel attack; however, this issue is outside the scope of our contribution. Hopefully, we will share our new findings soon.

Round 2

Reviewer 1 Report

My previous comments have been addressed in the revised version. I would like to recommend this paper be accepted for publication in the present form.

Reviewer 2 Report

The authors have satisfactorily modified their manuscript according to my previous criticisms. Therefore, I recommend the publication of this manuscript.